

# Impacts of abiotic factors and pesticide on the development, phenology, and reproductive biology of pink bollworm, *Pectinophora gossypiella* (Saunders) (Lepidoptera: Gelechiidae)

Muhammad Jalal Arif[1,*], Ahmad Nawaz[2,*], Muhammad Sufyan[1], Muhammad Dildar Gogi[1], Zain UlAbdin[1], Muhammad Tayyib[1], Abid Ali[1,3], Waqar Majeed[4], Manel Ben Ali[5] and Amor Hedfi[5]

[1] Deaprtment of Entomology, Faculty of Agriculture, University of Agriculture Faisalabad, Faisalabad, Punjab, Pakistan
[2] Department of Plant Sciences, College of Agricultural and Marine Sciences, Sultan Qaboos University, Muscat, Oman
[3] College of Life Science, Shenyang Normal University, Shenyang, Liaoning, China
[4] Department of Zoology, Wildlife and Fisheries, University of Agriculture Faisalabad, Faisalabad, Pakistan
[5] Department of Biology, College of Sciences, Taif University, Taif, Saudi Arabia
* These authors contributed equally to this work.

Corresponding authors
Ahmad Nawaz,
nawazrajpoot65@gmail.com
Muhammad Dildar Gogi,
drmdgogi1974@gmail.com

## ABSTRACT

The pink bollworm, *Pectinophora gossypiella* (Saunders) (Lepidoptera: Gelechiidae) is a serious insect pest of cotton crop. The studies to evaluate the impact of abiotic factors on cotton pests' biology are limited. The current study was undertaken to determine the impact of abiotic factors (temperature, humidity, photoperiod) and an insecticide (lambda-cyhalothrin) on the biological aspects of *P. gossypiella*. The results revealed that all the treatments showed a significant impact on different life parameters of *P. gossypiella*. The temperature exposure at 27 °C revealed a total life span of about 33 days. Maximum mortality for larvae (51.11%), pupae (59.04%) and adults (61.92%) were recorded at 33 °C exposure. Both low and high humidity levels caused negative impacts on life parameters of *P. gossypiella*. The life span was completed in about 30 days at 60% relative humidity (RH). Maximum mortality for larvae (75.00%) and pupae (49.59%) were recorded at 80% RH level exposure, while adult mortality was maximum (63.09%) at 40% RH level followed by 80% RH level (55.52%). The *P. gossypiella* exhibited a life span of about 32 days at 14:10 light-dark period. The larval mortality was maximum (14.83%) at 12:12 light-dark period while pupal (47.36%) and adult (48.84%) mortality was maximum at 16:08 light-dark period. Lambdacyhalothrin (LC) showed dose dependent negative impacts on biological aspects of *P. gossypiella*. The *P. gossypiella* exhibited a life span of about 26 days at 0.5 ppm LC concentration. The *P. gossypiella* exposure to highest concentration (LC) revealed maximum mortality of larval (80.22%), pupal (64.63%) and adult (70.74%) stages. Conclusively, the best suited abiotic factor ranges were 27 °C (temperature), 60% (RH) and 14:10 (light-dark) which can be used for successful rearing and bioassay activities of *P. gossypiella*.

# INTRODUCTION

Pink bollworm, *Pectinophora gossypiella* (Saunders) (Lepidoptera: Gelechiidae), belongs to the holometabola (egg, larva, pupa, and adult) group of insects. The female moth may lay 100–200 eggs after 2 days of pre-oviposition at the early stage of cotton bolls (*Gull-e-Fareen et al., 2021*). Eggs are lustrous white and oval shape measuring up to 0.5 mm (length) and 0.25 mm (width). Eggs are laid singly or in cluster form on green bolls, squares and flowers (*Madhu & Mohan, 2021*). The first instar larvae are white and turn pink as they mature. The larva has a distinctive dark brown head with prothoracic shield. The larvae bore into cotton bolls and feeds on the seed. The larval period lasts for 10–14 days (*Bhute et al., 2023*). Pupa measures approximately 7 mm in length with light brown to dark brown color. Pupation occurs for 7–8 days in the top layer of the soil or plant debris under the cotton plant. The grey, brown adult has black stripes on the fore wings and silver colored hind wings (*Sarwar, 2017*). In addition, *P. gossypiella* survives in four seasonal cycles (winter, spring, fall and summer). In winter cycle, it undergoes diapause in the form of full-grown 4th larval instar (*Parmar & Patel, 2016*). The spring cycle starts when diapausing larva begins to respond to temperature and moist conditions. The summer cycle lasts around one month period and *P. gossypiella* can pass through 1–4 summer generations depending on local circumstances (*Sarwar, 2017*). Some larvae begin to prepare for winter diapause in late August during the autumn cycle. Shorter days (reduced photoperiod) and temperatures (≤70 °F) are the primary causes of diapause (*Sarwar, 2017*). This diapause in the field increases dramatically after mid-September as days length begins to shorten.

Climate change is one of the greatest concerns and research challenges. The present carbon dioxide trajectory will raise global temperatures up to 4.4 °C by the end of the century (*IPCC, 2021*). Climate change affects insects directly by altering their physiological mechanisms, developmental rate and phenology (*Rudolf & Singh, 2013*; *Reuman, Holt & Yvon-Durocher, 2013*; *Amarasekare & Coutinho, 2014*). The temperature exposure at 29 °C shortens the incubation period of *P. gossypiella* eggs while coolest temperature (70 °F) with longer exposure of reared larvae causes delayed pupation in them (*Reda & El-Nemaky, 2008*; *Qasim et al., 2018*; *Chen et al., 2020*). The active period of *P. gossypiella* is extended throughout the year. The pest population is thought to be at its peak at around 37 °C and minimum at 40 °C in June (*Nagaraju, Mohan & Keerthi, 2024*). More attacks of *P. gossypiella* were found in September (*Verma et al., 2017*). Like temperature and humidity, photoperiod may also cause positive and negative impacts on the *P. gossypiella* population dynamics.

The *Pectinophora gossypiella* is a major insect pest of cotton damaging squares, flowers, and bolls. Larvae burrow into bolls to feed on seeds, that causes a significant loss in cotton output in terms of quantity and quality. It also causes boll rotting, premature or partial boll opening, decrease in staple length and strength which increases the amount of garbage in

the lint (*Sabry, Hassan & Rahman, 2014*; *Ahmed, 2020*). The larvae feed on cotton flowers and bolls causing 2.8% to 61.9% loss in seed cotton production, 2.1% to 47.10% reduction in oil content and 10.70% to 59.20% loss in normal boll opening (*Naik et al., 2018*). It is difficult to control with insecticides because the larval stage feeds inside cotton bolls. Therefore, insecticide treatment alone may not be effective in pest control. In addition, pesticides are hazardous to non-target organisms (*Nawaz et al., 2021*) and *P. gossypiella* has also developed resistance to several pesticides used in cotton (*Hany, Abd-Elhady & Abd El-Aal, 2011*; *Dhurua & Gujar, 2011*; *Naik et al., 2018*, *2023*; *Kumar & Grewal, 2023*). It has been identified presenting and gaining resistance to hazardous chemicals during each cotton growing season (*Hany, Abd-Elhady & Abd El-Aal, 2011*; *Sarwar & Sattar, 2016*). However, after more than four decades of use, pesticides have not eliminated the problem anywhere on the globe. Furthermore, the climatic factors can also affect the efficacy of pesticides (*Matzrafi, 2019*; *Mao et al., 2019*; *Lurwanu et al., 2021*). Therefore, reconnoitering insect pests and insecticides with climatic variables is important to develop environment-based forecasting models. This may also help to improve the long duration prediction of pests and insecticide application (*Xiao & Wu, 2019*). Therefore, it was hypothesized that temperature, relative humidity, and photoperiod affect the biological parameters of *P. gossypiella* and temperature may also affect the efficacy of insecticides. For this purpose, we have studied the impact of temperature, relative humidity, and photoperiod climatic variables on the biological aspects of pink bollworm. In addition, the impact of insecticide against *P. gossypiella* was also investigated under laboratory conditions. The research outcomes revealed quite important information regarding the positive and negative impacts of tested parameters which can be further investigated to develop predictive models and successful rearing of *P. gossypiella*.

## MATERIALS AND METHODS

### Insect collection, rearing and starter culture

The pink bollworm infested bolls were collected by ZigZag method of pest scouting at morning time during August–September 2021 from the cotton fields located at Entomological Research Farm of University of Agriculture Faisalabad (here after read as UAF), Pakistan (31°–44′N, 73°–06′E). The cotton bolls with small holes on their outer layer were identified with magnifying glass and half cut into pieces with the help of knife as described in *Ihsan et al. (2021)*. The larvae were identified by observing their morphological characters (shape, size, color) under a stereo microscope. The cotton bolls infested with larvae (Fig. 1) were shifted to glass cages ($60 \times 60 \times 60$ cm$^3$) under controlled environmental conditions ($27 \pm 2$ °C temperatures and $70 \pm 10\%$ relative humidity (RH)) that facilitated pupae development to adult emergence in Pink Bollworm Rearing Laboratory at the Department of Entomology, UAF. The controlled conditions were maintained through a thermostat control mechanism. The adult male to female moths based on gonad and anal pore (*Ramya, Mohan & Joshi, 2020*) at 1:1 was manually released in oviposition glass chimneys covered with towel tissue paper (white) as an egg laying substrate. A glass vial (height = 70 mm; external diameter = 21.5 mm; opening diameter = 12 mm) containing adult diet (deca vitamin drop: 1 mL and 10% honey

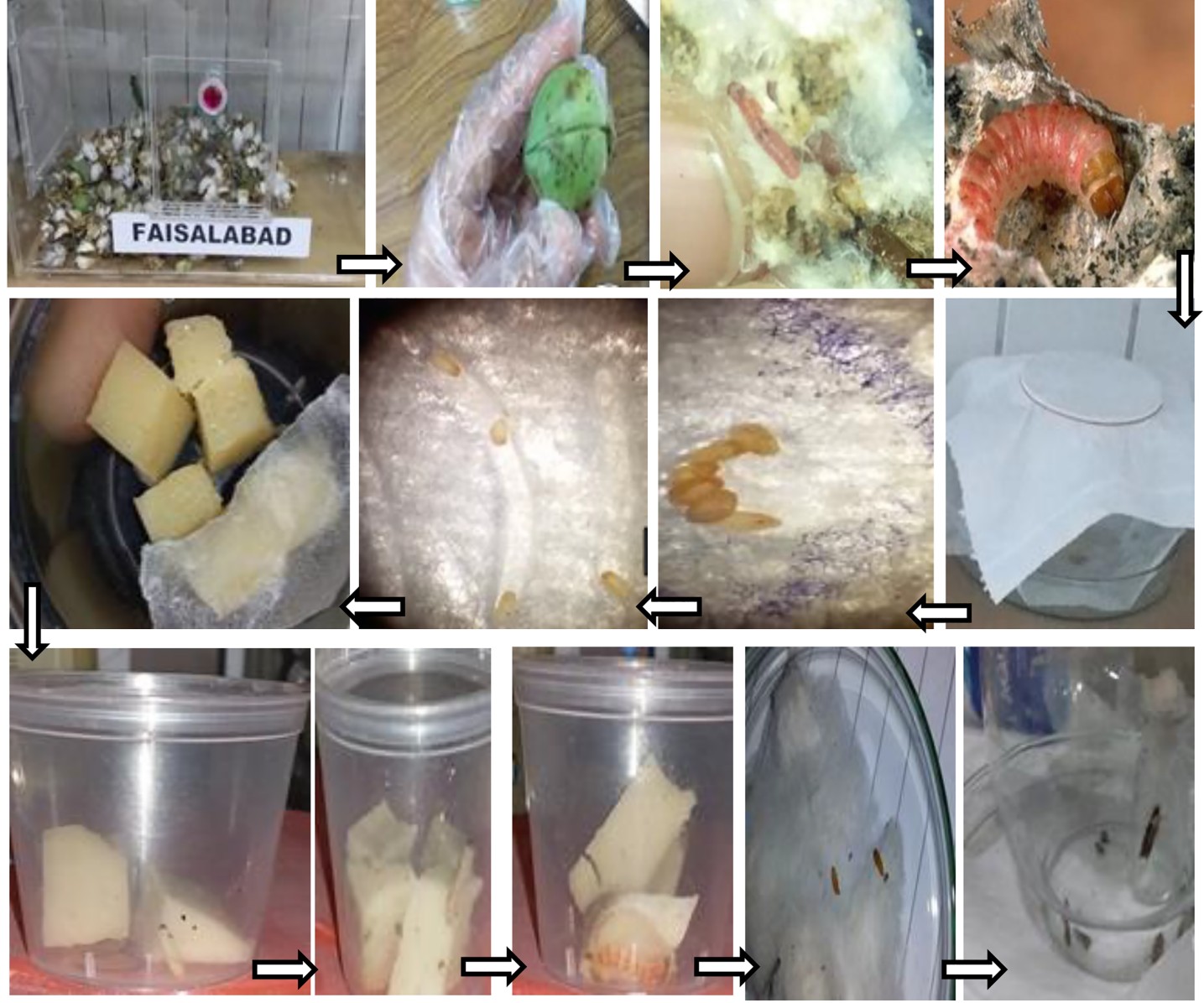

**Figure 1** **Rearing procedure of pink bollworm.** Complete procedure of collection of *P. gossypiella* infested cotton bolls from district Faisalabad following adult emergence and egg laying in glass chimney covered with oviposition substrate (groves) showing egg laid pattern in groves of towel tissue. The plastic cup with neonates shifted on diet and male and female dark brown pupae following PBW adult emergence (*Ihsan et al., 2021*).

solution) for egg laying purpose was also placed inside each glass chimney. After three days, the eggs were collected from the ventral side of tissue paper and shifted to plastic cups ($3.8 \times 3.4 \times 3$ cm$^3$) for hatching (Fig. 1). Pink bollworm eggs hatched in 2–3 days under controlled conditions as mentioned above. The duration of larval hatching was recorded before shifting neonates to rearing cups ($3.8 \times 3.4 \times 3$ cm$^3$). The plastic cups having neonates were exposed to torchlight that facilitated their transfer to rearing cups using a fine camel hairbrush (Fig. 1). The larval hatching was recorded daily to build up larval

culture for further rearing purposes. The hatched larvae were transferred to larval rearing cups ($3.8 \times 3.4 \times 3$ cm$^3$) containing larval diet. The neonate larvae were shifted to transparent larval rearing cups ($3.8 \times 3.4 \times 3$ cm$^3$) with lids (having small pores for air circulation) to prevent larval escape, predator entry, diet contamination, and dehydration. These larval rearing cups were provided with cubes of larval diet. Fine Camel's hairbrush was used to release freshly emerged neonates onto diet cubes @ 2 larvae/cup in 10 replications of each treatment. For successful larval development, larvae were shifted onto a fresh diet after every third day (Fig. 1). The 4th larvae were sexed, weighed, and counted separately for adult pairing. After successful pupation, the emerged adults were released in pairs into oviposition glass chimneys for mating and egg-laying purposes. Wide-mouthed round glass chimneys were used to prevent excessive flight activity, crowding of adults, preserve scales and to facilitate mating among adults.

## Experimental design

The experiment was set up in a completely randomized design (CRD) comprising of four treatments including temperature, relative humidity, photoperiod, and insecticide while each treatment was replicated 10 times with ten larvae per replication. The complete life stages of pink bollworm were studied under each treatment parameter. The standard diet (the wheat germ as the main ingredient), was prepared as described by *Akhtar et al. (2024)* as per diet composition. The previously described procedure (*Ihsan et al., 2021*) was used to avoid fungal contamination during diet preparation. The equipment was first sterilized with 5% ethanol solution (25 mL water) and then autoclaved at 250 °F (121 °C) and 15 psi for a prescribed time usually 30–60 min. The standard diet was based on the techniques suggested by *Akhtar et al. (2024)* and *Wu et al. (2008)*. All ingredients were accurately weighed using electronic balance (CE Model EJ-3238+) followed by three fractions (A, B and C) of ingredients, so a well-mixed product was obtained. Fraction A ingredients were stirred well in 230 mL of distilled water in a 1,000 mL measuring beaker. Then fraction B comprised of deca vitamins (0.0l mL) was separately mixed in 10 mL of water in a measuring cylinder to make vitamin solution. Fraction C comprised of agar as a thickening agent and was separately well stirred in 500 mL of distilled water in a 1,000 mL beaker. All fractions were blended with the addition of 3.3 mL corn oil and 2 mL honey into a blender mixture. Finally, poured into the Petri dishes ($150 \times 15$ mm) and allowed to solidify for 10 min. Once the diet in petri dishes solidified, it was cut with a spatula into small diet cubes (¼ inches) and placed in transparent larval rearing plastic cups (3–4/cup). The biological parameters of pink bollworm were studied under five different but constant temperatures (21, 24, 27, 30 and 33 °C) with constant relative humidity (65 ± 5%). The adult moths (male and female) were subjected to each temperature and the longevity of adults, fecundity of females, egg incubation period, hatching percentage, complete larval and pupal periods were studies in incubator (Incubator IN/INplus series IN series models, 32 l) that were set to each different range of temperature separately. To study the impact of relative humidity (RH), five different ranges of RH were selected as 40%, 50%, 60%, 70% and 80% RH. A humidifier (black decker) was used to maintain the different levels of

relative humidity. The RH was monitored with the help of humidity sensor. Each parameter of pink bollworm biology was observed independently under the selected levels of humidity percentages. The data about the implications of very low and high RH (%) was recorded for the statistical analysis. Similarly, the impact of photoperiod was investigated with five different lengths of light and darkness ranges such as 12:12, 13:11, 14:10, 15:9 and 16:8 L:D. The photoperiod was also selected to check the stress on different parameters of pink bollworm biology. In addition, another bioassay experiment was conducted to evaluate the stress of chemicals against different life stages of pink bollworm and pyrethroids group product lambda-cyhalothrin (2.5%; Syngenta®) was selected for the trial due to its excessive use in cotton against bollworms. Five different concentrations of lambda-cyhalothrin (0.5, 1, 1.5, 2 and 2.5 ppm) were prepared. Each concentration of lambda-cyhalothrin was prepared in 100 mL of distilled water and was applied to different stages of pink bollworm using a hand sprayer (WIRELESS ATOMIZER SPRAYER, A7-01). The changes in the biological parameters were noticed and data was collected to apply statistical analysis.

## Statistical analyses

The data recorded against different biological parameters under different treatments were subjected to data transformation using $\sqrt{(x + 0.5)}$ to achieve normality before analysis; however, untransformed means were presented in the figures. The biological parameters of *P. gossypiella* was analysed using a generalized linear model (*Tabachnick & Fidell, 2001*) through one way analysis of variance (ANOVA) to determine the parameters of significance and mean values for different treatments. The means of significant treatments were compared with Tukey's honestly significant difference at the 5% probability level, as performed by *Danho, Gaspar & Haubruge (2002)*. The statistical analysis was performed using Statistica 8.1 software.

# RESULTS

## Temperature impact on biology

Temperature exerted a very significant impact on the biological parameters of *P. gossypiella* (Fig. 2). The egg hatching gradually increased with increase in temperature. Maximum egg hatching (63.66) was recorded at 27 °C and decreased with an increase in temperature exposure. The incubation period of eggs showed varied responses to temperature. The larval development was also temperature dependent. The larvae exposed to high (30, 33 °C) or low temperature (21, 24 °C) showed longer development period (14–17 days). The minimum time for larval development was recorded at 27 °C (10.83 days). The maximum number of larvae goes to diapause at 21 °C (43.52%) followed by 23 °C (28.99%), 30 °C (21.37%) and 33 °C (15.89%). Minimum larvae undergo diapause at 27 °C (11.51%). Similar trends were recorded for larval weight and larval mortality. Minimum mortality (16.25%) was at 27 °C while highest mortality (51.11%) was recorded at 33 °C followed 21 °C (39.76%). Subsequently, the pupal recovery was significantly higher (82.70%) at 27 °C and gradually decreased with an increase in temperature (39.56%). A similar trend was recorded during the decrease in temperature (20.91%). Like the larval

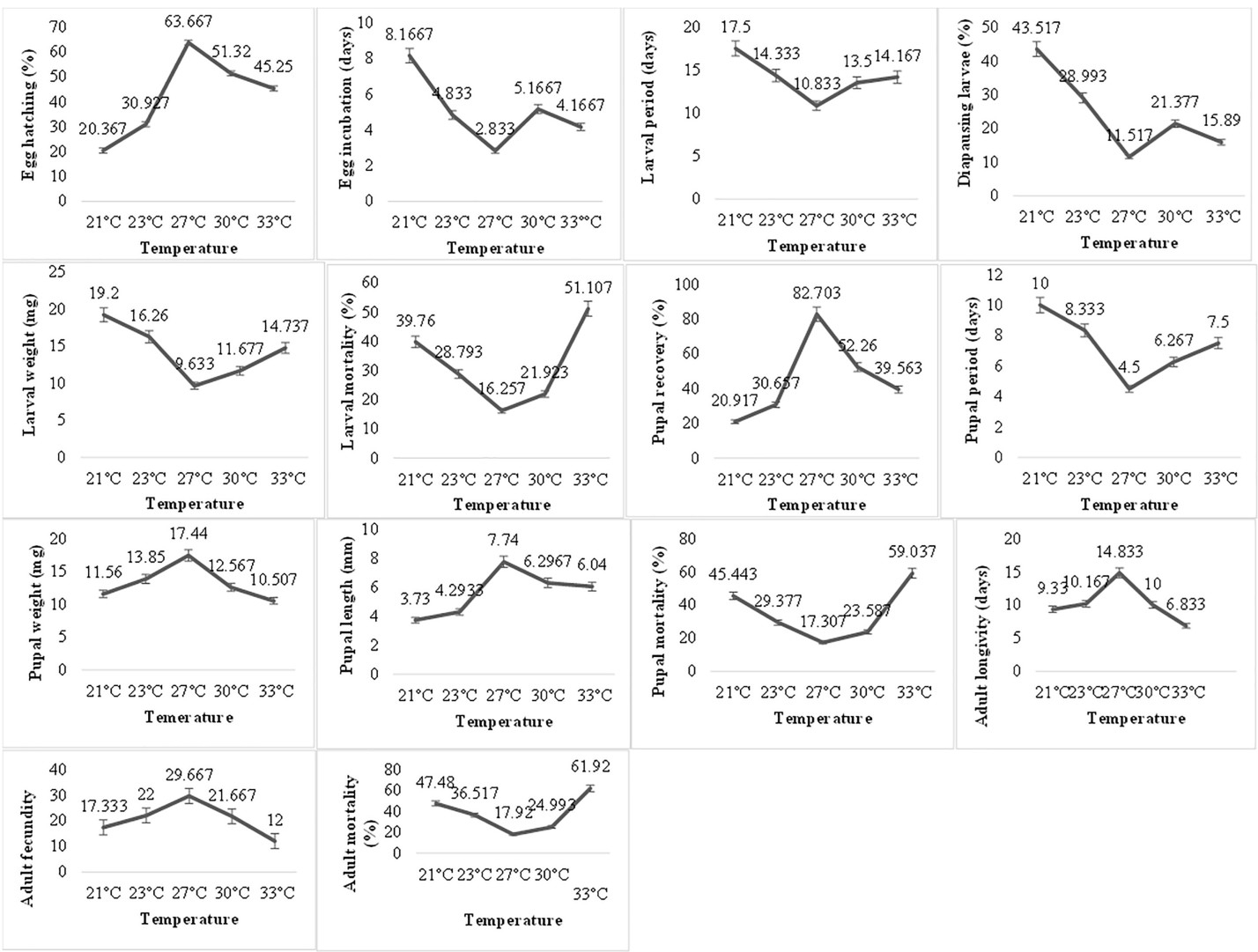

**Figure 2 Temperature impact on the biology of pink bollworm.** Impact of different temperature regimes on life parameters of pink bollworm showing significantly variable responses at all stages of life. The analysis revealed 27 °C as the most suitable temperature for the development of pink bollworm on all biological parameters. The exposure of high or low temperature significantly affecting the development of pink bollworm.

period, the pupal period was also very low (4.5 days) at 27 °C, while maximum pupal period (10 days) was recorded at 21 °C. In contrast to larval weight, the maximum pupal weight (17.44 mg) was recorded at 27 °C showing an average length of 7.74 mm. The pupal mortality (59.03%) was also higher at maximum temperature exposure while minimum at 27 °C (17.307%). The optimum temperature exposure also exhibited maximum adult longevity (18.83 days) and fecundity (29.66 eggs). The maximum temperature exposure (33 °C) caused maximum adult mortality of 61.92% followed 47.48% mortality at lowest temperature (21 °C) exposure. Overall, the variation in temperature exposure significantly altered the biological parameters of *P. gossypiella* and the most suitable temperature was 27 °C.

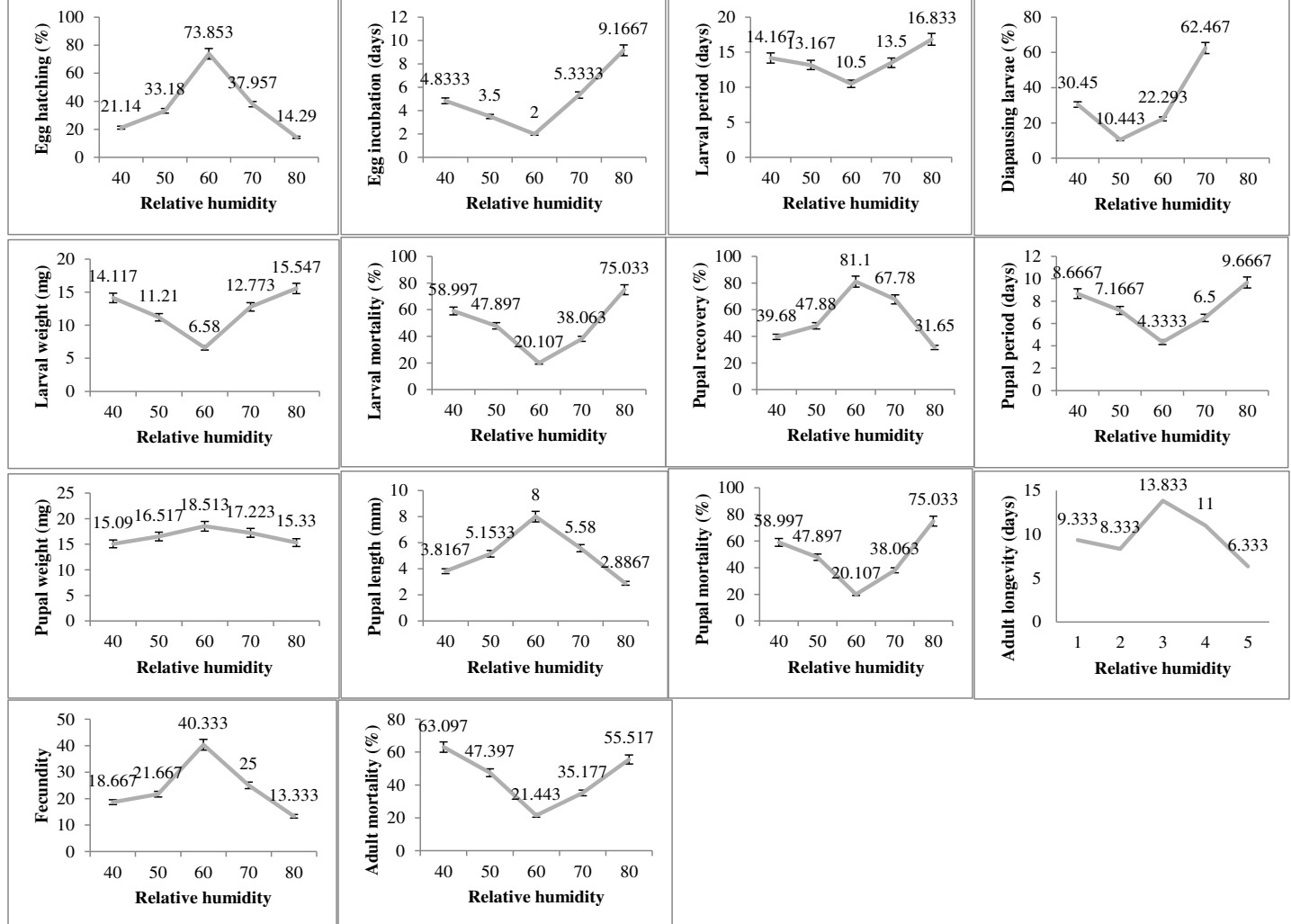

**Figure 3 Impact of relative humidity on the biology of pink bollworm.** Impact of different levels of relative humidity on life parameters of pink bollworm showing significantly variable responses at all stages of life. The exposure to 60% relative humidity level significantly exerting positive impact on the biological development of pink bollworm. Very high or low level of relative humidity causing negative impacts.

## Humidity impact on biology

The humidity exposure to *P. gossypiella* exhibited varied impacts on different biological parameters and the most suitable humidity level was recorded as 60% RH (Fig. 3). The egg hatching was significantly higher (73.85%) at 60% humidity level with minimum incubation period (2 days). The minimum egg hatching (14.29%) was recorded at 80% RH. The time for larval period was also minimum (10 days) at 60% humidity. The maximum larval period was 16.83 days at 80% humidity level followed by 40% RH with 14.16 days of larval period. Similarly, the diapausing larvae were also minimum (10.44%) during 60% humidity exposure. The deviation from 60% humidity level significantly increased or decreased the larval weight and larval mortality. The maximum larval weight and larval mortality was recorded as 15.54 mg and 75.03%, respectively at 80% humidity level. The minimum larval mortality was 20.10% at 60% humidity level. The maximum pupal

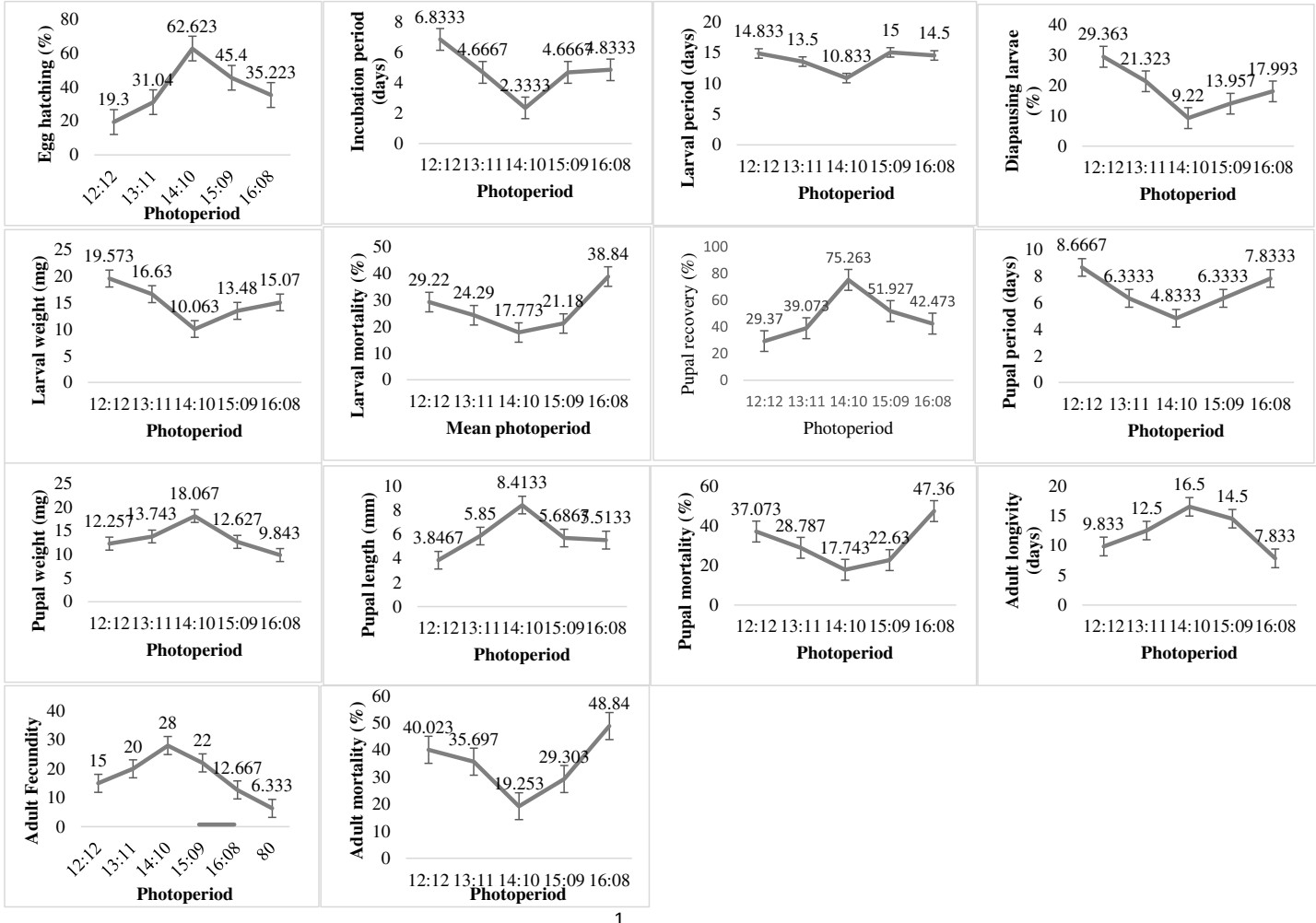

**Figure 4 Impact of photoperiod on the biology of pink bollworm.** Impact of different levels of photoperiod on life parameters of pink bollworm showing significantly variable responses at all stages of life. The analysis of data revealed that the 14:10 light and dark photoperiod exposure is significantly more suitable for better development of pink bollworm showing minimum larval, pupal and adult mortality.

recovery (81.1%), pupal weight (18.51) and pupal length (8 mm) were recorded at 60% humidity exposure level. The maximum pupal period (9.66 days) and pupal mortality (75.03%) was recorded at highest humidity exposure level (80%). The adult average survival was maximum 13.83 days at 60% humidity exposure level, while minimum survival 6.33 days was recorded at 80% humidity exposure level.

## Photoperiod impact on biology

The variation in photoperiod exposure times significantly affected the biological parameters of the *P. gossypiella* (Fig. 4). The minimum incubation period and maximum hatching was 2.33 days and 62.62% respectively at photoperiod 14:10. The 12 h dark and light exposure revealed minimum egg hatching (19.3%) of *P. gossypiella*. The larval development was also significantly affected by photoperiod variations. The minimum diapausing larvae (9.22%), larval period (10.83 days), larval weight (10.06 mg) and larval

mortality (17.77%) was recorded at 14:10 light-dark period. While the maximum diapausing larvae (20.36%), larval period (14.83 days), and larval weight (19.53 mg) was recorded at 12:12 light-dark period. The larval mortality was maximum (38.84%) at 16:08 light-dark period. The maximum (75.26%) and minimum (29.37%) pupal recovery recorded at 14:10 and 12:12 light-dark periods respectively. Like larval parameters, the minimum pupal period (4.83 days), pupal mortality (17.74%) and maximum pupal weight (18.06 mg) as well as pupal length (8.41 mm) was observed during 14:10 light-dark period exposure. Like immature stage, the *P. gossypiella* adults also showed varied responses to photoperiod exposure. The minimum and maximum adult longevity (7.83, 16.5 days) and fecundity (6.33/female, 28/female) was recorded at 16:08 and 14:10 photoperiods exposure respectively.

## Pesticide impact on biology

In contrast to the impact of abiotic factor showing varied responses at different levels of exposure, the pesticide lambda-cyhalothrin (LC) exhibited dose dependent impact on all the studied biological parameters of *P. gossypiella* (Fig. 5). The egg hatching was significantly higher at lowest concentration (68.96%) exposure and gradually decreased up to only 11.3% with increase in concentration exposure. The egg incubation period was also increased (4.83–9.5 days) with increased concentration of pesticide. A similar trend was recorded for larval biological parameters (larval period, diapause, weight, mortality). The increase in LC concentration increased the larval period from 4.5 to 12 days, the diapausing larvae from 9.03 to 54.22%, larval weight from 8.05 to 17.04 mg and larval mortality from 17.48 to 80.21%. The pupal recovery declined from 80.73 to 24.07% from lower to higher concentration exposure respectively. Like the larval period, the pupal period was also increased with increased concentrations of LC. The pupal weight and pupal length were decreased while larval mortality was maximum (64.62%) at the highest concentration of LC exposure to *P. gossypiella*. Adult *P. gossypiella* was also proved susceptible to LC and showed a decrease in its average longevity from 14.5 days to 2.33 days and fecundity from 24 eggs per female to only 5 eggs per female from lower to higher LC concentrations respectively. Adult mortality increased with increasing LC concentrations and maximum mortality was 70.74% at highest concentration used.

## DISCUSSION

*Pectinophora gossypiella* is a multivoltine herbivore insect pest of cotton and other closely related species. The population dynamics, establishment, and biology of insects strongly influenced by abiotic factors especially temperature and humidity (*Jaworski & Hilszczanski, 2013*; *Nagaraju, Mohan & Keerthi, 2024*). The present study was conducted to check the temperature, relative humidity, photoperiod, and insecticide impacts on the biological parameters of *P. gossypiella* under controlled conditions. The present findings showed a varied relationship between biological parameters of pink bollworm and the abiotic factors especially the temperature ranges 21–33 °C. Empirical data on biological parameters of *P. gossypiella* are particularly valuable when looking for various area wide

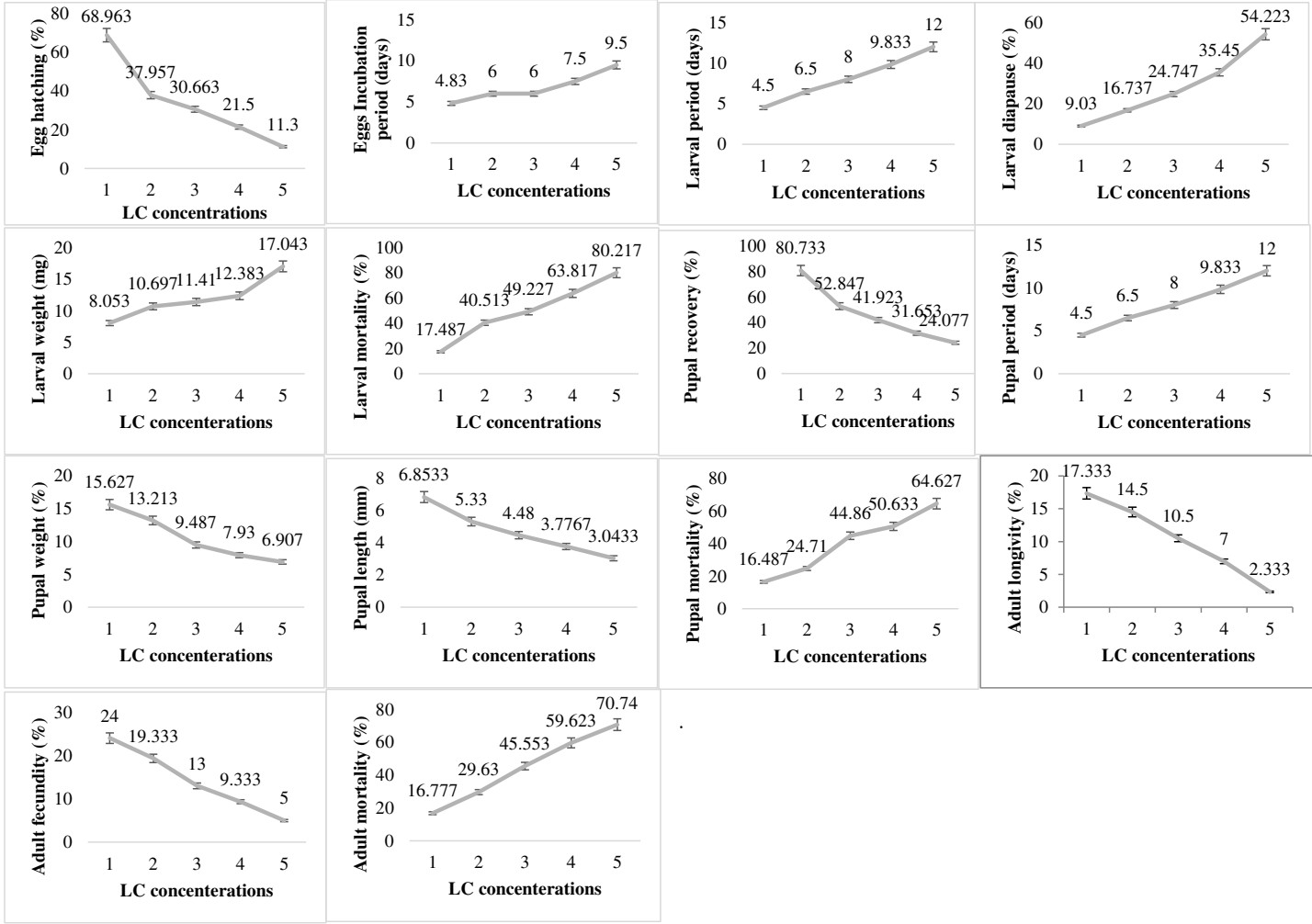

**Figure 5 Impact of pesticide lambda cyhalothrin on the biology of pink bollworm.** Impact of different concentrations of Lambda cyhalothrin on life parameters of pink bollworm showing significantly variable responses at all stages of life. The lambda cyhalothrin exerted dose dependent exposure on all the biological parameters of pink bollworm. Interesting the maximum egg hatching was recorded at lowest sublethal dose.

control options. The *P. gossypiella* showed highest adult longevity (14.83 days) at 27 ± 1 °C as compared to other ranges and the lowest longevity were recorded (6.83 days) at high temperature 33 ± 1 °C. Similar findings were recorded by *Peddu et al. (2020)* who conducted an experiment to check the adult longevity at temperature range between 15 and 35 °C. In another experiment adult longevity of pink bollworm decreased significantly with an increase in temperature (*Hussain et al., 2023*). Similarly, the adult fecundity (29.66) and egg hatching (63.66%) rate were recorded significantly higher at 27 °C. The current study agreement with *Peddu et al. (2020)* who reported that the shortened and reduced fecundity rate were recorded at lower temperature as compared to higher temperature range. Similarly, the pre- and post-oviposition rates were also recorded by *El-Lebody, Mostafa & Rizk (2015)* and *Zinzuvadiya et al. (2017)* as maximum eggs laying (102.6 eggs per female) were recorded at temperature range 25 °C and decreased

considerably at lower and higher temperature 20 and 30 °C, respectively. However, the fecundity is also influenced by various other factors like nutrition, relative humidity, food availability and light intensity *etc.* (*Attique et al., 2004*). *Nagaraju, Mohan & Keerthi (2024)* reported 43–45 days the life cycle of *P. gossypiella* at 20 °C which is also in agreement with current findings of 40–43 days life cycle at 21 °C. The total life span of *P. gossypiella* was around 31–33 days at maximum temperature exposure of 33 °C which could reduce more with an increase in temperature range (*Nagaraju, Mohan & Keerthi, 2024*). The highest adult mean mortality was recoded (61.92%) at 33 °C which is significantly greater as compared to other temperature ranges. Similarly, the egg hatching was also increased with an increase in temperature but up to a certain limit. The egg incubation period was 8.16 days at 21 °C and lowest was 2.83 days at 27 °C. The current findings agree with *Philipp & Watson (1971)* stated that fecundity rate and egg hatching of *P. gossypiella* significantly reduced at higher temperature. The pre-oviposition period was longest at a fluctuating temperature of 58°–91 °F and was shorter and more constant at the other temperatures. These findings reconfirm the fact that temperature is one of the key lifespan determinants in insects (*Garcia et al., 2017*; *Du Plessis, Schlemmer & Van den Berg, 2020*; *Mołon et al., 2020*; *Díaz-Álvarez et al., 2021*).

Similarly, larval diapause is high at low temperature 21 °C which goes on decreasing with increase in temperature up to 27 °C. The mean value of diapause is lowest at 27 °C and larva is more active. The increase in temperature from 27 °C also increased the larval diapause and *vice versa*. The current study agrees with *Pooja et al. (2022)* who reported that the larval and pupal period decreased at high temperature and constant at moderate temperature range. Likewise, it has been also reported that the larval and pupal weight is greater at moderate temperature and reduced at climatic extremes (*Hussain et al., 2023*). Similarly, *Raina & Bell (1974)* reported that South Indian pink bollworm failed to entered diapause while from Arizona enter the diapause at 19 °C. However, present results are deviating from the findings of *Kiranamaya et al. (2020)* who quoted that diapause inducing daily temperature and photoperiod were 20 °C and 12 h, respectively which depends upon hibernaculum sites. Overall change in temperature, humidity and photoperiod can affect both physiology and behavior of insects and plants. Moreover, a slight increase in temperature can dramatically affect the energy source of insects in diapause and could affect all metamorphic stages through increased respiration, decreased growth, reproduction and even survival (*Hahn & Denlinger, 2011*).

The humidity and photoperiod are also important factors for the development of arthropods (*Bale & Hayward, 2010*; *He et al., 2021*) but the data regarding the impact of humidity and photoperiod regarding *P. gossypiella* is lacking. The current finding exhibited varied impacts on different biological parameters with the most suitable humidity level of 60% RH (Fig. 3). There were significant impacts of humidity exposure levels and the developmental parameters of *P. gossypiella*. The minimum and maximum egg hatching was 14.29% and 73.85% at 80% and 60% RH. The tole life span of *P. gossypiella* was around 42 days at 80% RH followed 37 days (40% RH), 36 days (70% RH), 34 days (50% RH) and 30 days (60% RH). It is evident that relative humidity can influence insect development

(*Winkler et al., 2020*) and the combined impact of humidity with heat stress can reduce the life span of insects (*Bubliy et al., 2012*). *He et al. (2021)* reported the higher pupal survival and adult emergence rate at the higher RH. This statement is contradictory to the current finding where maximum 75.03% pupal mortality was recorded at maximum RH (80%). But it has been also reported that high humidity levels are very conducive for several insects *e.g.*, *Trichogramma dendrolimi*, *Carposina niponensis*, and *Araecerus fasciculatus* (*Li et al., 2006*, *Yang et al., 2016*, *Zhu et al., 2016*). Therefore, the increase in environmental temperature can pose a serious concern about pest infestation with reduced life cycle. The life cycle span may decrease, and pest infestation may increase. It has been reported that alterations in ambient humidity can directly affect insect development and metabolism *e.g.*, high or low RH often delays the development of eggs and larvae of different lepidoptera (*Heliothis virescens* Fabricius; *Helicoverpa armigera*; *Leucania separata* Walker) insect pests (*He et al., 2021*). The variation in photoperiod exposure times significantly affected the biological parameters of the *P. gossypiella* (Fig. 4). The tole life span of *P. gossypiella* was around 40 days at 15:09 photoperiod followed 39, 35, 34, and 33 days at 12:12, 16:08, 14:10 and 13:11 photoperiod respectively. Many studies reported that increasing the light ratio to dark intricate the reduction in diapause in *P. gossypiella* (*Kaltsa, Milonas & Savopoulou-Soultani, 2006*; *Mohapatra, 2007*). Therefore, the addition of variation in the photoperiod response (*versus Grevstad & Coop, 2015*) offers improved accuracy in determining the voltinism and mismatch estimation in the targeted population. This can help to develop prediction models of important insect pests.

There are no detailed studies regarding the impacts of lambda-cyhalothrin on the biology of *P. gossypiella*. Most of the studies are done to highlight its egg and larval toxicity against in *P. gossypiella* (*Radwan et al., 2018*; *Chaudhari, Panickar & Chandaragi, 2023*; *Busnoor et al., 2024*). In current finding, lambda-cyhalothrn exhibited dose dependent effect against target insect pest. The higher concentration (2.5 ppm) significantly reduced the fecundity rate, hatching percentage, incubation period, larval diapause, larval weight, larval period, larval mortality, pupal recovery, pupal period, pupal mortality, pupal weight, adult longevity, and adult mortality as compared to lower doses. These findings agree with *Abbas et al. (2017)* who concluded that the application of lambda-cyhalothrin cause's higher mortality (56.07%) as compared to indoxicarb, masrona and emamectin, and significantly prolonged larval and pupal periods, longevity, and reduced fecundity rate of *P. gossypiella*. *Sabry (2013)* also reported significant variations on the susceptibility of *P. gossypiella* among buprofezin and lambda-cyhalothrin at different concentration while there was no discernible distinction between thiamethoxam and lambda-cyhalothrin induced toxicity. Moreover, *Abd EL-Mohsen et al. (2013)* recommended that use of lambda-cyhalothrin decreases the infestation rate of *P. gossypiella* of about 38.02% and was highly effective against larvae. Likewise, *Fouda et al. (2017)* proposed that the lambda-cyhalothrin at its lower concentration $LC_{50}$ causes 9.68% mortality and at higher $LC_{95}$ causes 65.5% mortality to *P. gossypiella* larvae in field condition. *Radwan & El-Malla (2010)* were also considered lambda-cyhalothrin as the most impactful insecticide against the adult moth of *P. gossypiella* in field conditions and build up low resistance. Similar

findings were also reported in other studies (*Khan et al., 2007*; *Balakrishman, Kumar & Sivasubramanan, 2009*) that the lambda-chylothrin, bifenthrin and cypermethrin provide effective control by decreasing the bollworm incidence in cotton crop, at recommended doses (*Khattak et al., 2004*). *Sabry (2013)* ranked buprofezin more effective than lambda-chylothrin and thiamethoxam. In addition, the use of biopesticides or integration of insecticides with biological agents could improve the efficacy of pesticides and reduce the concerns of resistance development in the target insect pests (*Nawaz et al., 2020*, *2022*).

Conclusively, it is evident that climate changes are occurring very rapidly and the response of insect pests to these changes could be variable. The current investigation regarding the impact of temperature, humidity, photoperiod and pesticide revealed varied impacts on the biological parameters of *P. gossypiella*. Therefore, sustainable management of *P. gossypiella* is highly desirable especially the development of area wide management programs which needs a detailed study about the pest and its correlation with environmental factors. Current findings provide a baseline for researchers and other stakeholders (extension personnel, agricultural producers, industry, and government administrators) to consider the changing environmental condition for the effective management of this serious pest. Taking the data on biology into account of current study can provide an element within integrated pest management system. The current findings will also be useful in the planning, implementation, and evaluation of management strategies. However still needed further field trials for confirmation and better understanding to manage this insect pest. Therefore, it is recommended for future studies to investigate the pesticide impacts on the biological parameters with fluctuation abiotic factors (temperature, humidity, photoperiod) for a precise understanding of their possible implication in changing climatic conditions.

### Funding
This work was funded by the Punjab Agriculture Research Board (PARB) Project No. 888 and Taif University, Saudi Arabia, Project No. (TU-DSPP-2024-158). The funders had no role in study design, data collection and analysis, decision to publish, or preparation of the manuscript.

### Grant Disclosures
The following grant information was disclosed by the authors:
Punjab Agriculture Research Board (PARB): 888.
Taif University, Saudi Arabia: TU-DSPP-2024-158.

### Competing Interests
The authors declare that they have no competing interests.

### Author Contributions
- Muhammad Jalal Arif conceived and designed the experiments, prepared figures and/or tables, and approved the final draft.

- Ahmad Nawaz conceived and designed the experiments, analyzed the data, prepared figures and/or tables, and approved the final draft.
- Muhammad Sufyan conceived and designed the experiments, prepared figures and/or tables, and approved the final draft.
- Muhammad Dildar Gogi performed the experiments, analyzed the data, authored or reviewed drafts of the article, and approved the final draft.
- Zain UlAbdin performed the experiments, authored or reviewed drafts of the article, and approved the final draft.
- Muhammad Tayyib performed the experiments, prepared figures and/or tables, and approved the final draft.
- Abid Ali analyzed the data, prepared figures and/or tables, and approved the final draft.
- Waqar Majeed performed the experiments, prepared figures and/or tables, and approved the final draft.
- Manel Ben Ali analyzed the data, authored or reviewed drafts of the article, and approved the final draft.
- Amor Hedfi analyzed the data, authored or reviewed drafts of the article, and approved the final draft.

## Field Study Permissions

The following information was supplied relating to field study approvals (*i.e.*, approving body and any reference numbers):

The cotton boll samples were collected from the farmer fields.

## Data Availability

The raw data are available in the Supplemental Files.

## Supplemental Information

Supplemental information for this article can be found online at http://dx.doi.org/10.7717/peerj.18399#supplemental-information.

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
