# Peer review of "Impacts of abiotic factors and pesticide on the development, phenology, and reproductive biology of pink bollworm, Pectinophora gossypiella (Saunders) (Lepidoptera: Gelechiidae)"

_PeerJ, doi:10.7717/peerj.18399_

## Round 0.1 · original submission · Major Revisions

Dear author,

The pink boll worm is a serious insect pests of cotton and your study is important in understanding its biology in attacking the host crop. However, the manuscript needs to be reviewed by a professional language editor before re-submission.

the journal guidelines should be followed in processing the manuscript and especially the reference section. The interpretation of data in results and discussion section should be enhanced.

The valuable comments addressed by the reviewers should be followed, especially Reviewer 2 and 3.

Therefore, your manuscript needs major revision and re-submit after addressing the suggested comments.

**Language Note:** The review process has identified that the English language must be improved. PeerJ can provide language editing services - please contact us at [email protected] for pricing (be sure to provide your manuscript number and title). Alternatively, you should make your own arrangements to improve the language quality and provide details in your response letter. – PeerJ Staff

Reviewer 1 ·

Basic reporting

Original submission
Recommendation: major revision

Overview and general recommendations
It is a good study. But there are some points need to correct in review file

Experimental design

good

Validity of the findings

good

Annotated reviews are not available for download in order to protect the identity of reviewers who chose to remain anonymous.

·

Basic reporting

1. Language and Grammar: The manuscript contains minor grammatical errors and awkward phrasings. Thorough proofreading or language editing is recommended.
2. Consistency: Ensure consistency in the use of terms (e.g., "larval period" vs. "larval duration") throughout the manuscript.

Experimental design

1. Insect Collection: Provide more details about the collection process and criteria for selecting infested cotton bolls.
2. Control Conditions: The section mentions controlled environmental conditions but lacks specific details on how these conditions were maintained and monitored.
3. Replications: Based on preliminary analysis, is the sample size of ten larvae per replication adequate?
4. Diet Preparation: Streamline the description of diet preparation to focus on critical points.

Validity of the findings

1. Data Presentation: Incorporate more graphical representations (e.g., figures and tables) to enhance clarity.
2. Statistical Analysis: Include detailed statistical values (e.g., p-values, confidence intervals) for each significant result.
3. Comparative Analysis: To contextualize the findings, provide more comparisons with existing studies.
4. Interpretation of Results: Interpret the results more deeply, explaining the significance and mechanisms behind the observed effects.
5. Limitations: Acknowledge any potential weaknesses or confounding factors in the study.
6. Future Research: Provide specific recommendations for future research directions.
7. Summary of Findings: Make the conclusion succinct and directly tied to the study's objectives and hypotheses.
8. Practical Implications: Elaborate on the practical implications of the findings for cotton pest management and agricultural practices.

Additional comments

1. Formatting: Ensure all references are formatted consistently according to the journal’s guidelines.
2. Completeness: Make sure all citations are up-to-date and relevant.

Reviewer 3 ·

Basic reporting

The language is very poor. Authors must seek professional editing services before submission. So many ambiguities in the methodology and presenting results. Authors need to go through some standard articles and discuss them thoroughly.

Experimental design

The authors did not present their experimental design scientifically. The language should be improved.

Validity of the findings

The findings need to be relooked before making interpretations.

Annotated reviews are not available for download in order to protect the identity of reviewers who chose to remain anonymous.

---

## Round 0.2 · Minor Revisions

Dear authors

Thank you for addressing all of the reviewers' comments. A few more corrections need to be considered in a revised manuscript. These changes should lead to a much-strengthened publication.

Reviewer 1 ·

Basic reporting

accept

Experimental design

good

Validity of the findings

good

Reviewer 3 ·

Basic reporting

The manuscript improved a lot. Still, I could find some ambiguous and repeated sentences in most of the sections.

Experimental design

Need to be rearranged.

Validity of the findings

Recheck the values and discuss properly.

Additional comments

L- 12: Low and high humidity levels → Both low and high humidity caused
L 20: The P. gossypiella exhibited a life span of about 26 days at lowest concentration (0.5 ppm) → Sublethal effects may not reduce the life span (in fact, it extends in most species). Please check.
L 43: P. gossypiella to Pectinophora gossypiella
L 49: 4.4 0C → It is very high, Please check.
L96: The adult male to female moths based on gonad and anal pore → It is in the larval stage not in adults.
L154-155: The chemical treatment should be uniform and cannot be done in different stages. The bioassay is always performed during the larval stage. Please check and rewrite.
L156: Hand sprayer specifications? Company, model, etc.
L 161: The data on these biological parameters were can be rewritten as “The biological parameters of P. gossypiella was analysed
L 163: Delete from “technique……………… mean values for different tretaments”.
L 165: Delete “Whole of”
L 170-171: “The egg hatching gradually increased with an increase in temperature. Maximum egg hatching (63.66) was recorded at 27°C and decreased with an increase in temperature exposure” → Why this contrasting sentence was written in the results.? When you say egg hatching increases with an increase in temperature, then how it could decreased when temperature increases.? Authors should seriously look into this.
L173: “The larvae exposed to very high or low temperature showed longer development period (14-17 days)” → Again ambiguous statement. Longer development of larvae is because of high temperature or low temperature.? Say it properly.
L 174: The minimum larval development is always associated with higher temperature. When you have exposed the larvae up to 33 degrees, then how it was minimum in 27 degrees. Is there any basis for that.?
L 175-176: When the diapause is maximum at 21 degrees, then it is quite minimum at 33 degrees. How come it is maximum at 33 degrees and minimum at 27 degrees.? What do authors want to highlight here.? It seems the author wanted to project that 27 degrees is doing well for P. gossypiella by eliminating all other factors!!
L 195: Give space in between “The maximum”
L 198: Be specific, do not write “increased or decreased”. (It is ambiguous)
Check the manuscript with expertise and correct some grammatical mistakes. The discussion part could also be improved.

---

## Round 0.3 · accepted · Accept

All the minor revisions requested have been addressed. Your manuscript can now be accepted for publication in PeerJ. Congratulations.